# 'Voice needs teeth to have bite'! Expanding community-led multisectoral action-learning to address alcohol and drug abuse in rural South Africa

Lucia D'Ambruoso[1,2,3,4]*, Denny Mabetha[1,2], Rhian Twine[2], Maria van der Merwe[1,2,5], Jennifer Hove[1,2], Gerhard Goosen[6], Jerry Sigudla[6], Sophie Witter[7], On behalf of the Verbal Autopsy with Participatory Action Research (VAPAR)/Wits/Mpumalanga Department of Health Learning Platform

1 Aberdeen Centre for Health Data Science, Institute of Applied Health Sciences, School of Medicine, Medical Sciences and Nutrition, University of Aberdeen, Scotland, United Kingdom, 2 MRC/Wits Rural Public Health and Health Transitions Research Unit (Agincourt), School of Public Health, University of the Witwatersrand Johannesburg, South Africa, 3 Department of Epidemiology and Global Health, Umeå University, Sweden, 4 Public Health, National Health Service (NHS) Grampian, Scotland, United Kingdom, 5 Maria van der Merwe Consulting, White River, South Africa, 6 Mpumalanga Department of Health, Mbombela, South Africa, 7 Institute for Global Health and Development, Queen Margaret University, Scotland, United Kingdom

* lucia.dambruoso@abdn.ac.uk

**Data Availability Statement:** Data are available from the research team upon reasonable request. A non-author point of contact to field future requests

## Abstract

There is limited operational understanding of multisectoral action in health inclusive of communities as active change agents. The objectives were to: (a) develop community-led action-learning, advancing multisectoral responses for local public health problems; and (b) derive transferrable learning. Participants representing communities, government departments and non-governmental organisations in a rural district in South Africa co-designed the process. Participants identified and problematised local health concerns, coproduced and collectively analysed data, developed and implemented local action, and reflected on and refined the process. Project data were analysed to understand how to expand community-led action across sectors. Community actors identified alcohol and other drug (AOD) abuse as a major problem locally, and generated evidence depicting a self-sustaining problem, destructive of communities and disproportionately affecting children and young people. Community and government actors then developed action plans to rebuild community control over AOD harms. Implementation underscored community commitment, but also revealed organisational challenges and highlighted the importance of coordination with government reforms. While the action plan was only partially achieved, new relationships and collective capabilities were built, and the process was recommended for integration into district health planning and review. We created spaces engaging otherwise disconnected stakeholders to build dialogue, evidence, and action. Engagement needed time, space, and a sensitive, inclusive approach. Regular engagement helped develop collaborative mindsets. Credible, actionable information supported engagement. Collectively reflecting on and adapting the process supported aligning to local systems priorities and enabled uptake. The

(where authors are not available) is achds@abdn.ac.uk. Public deposition of the dataset would be in breach of the data management plan (DMP) within the study protocol approved by the research ethics boards in South Africa, the UK, and the permission for the study granted by the provincial health research committee, as well as the DMP within the funding proposal and the conditions upon which the research funding was granted.

**Funding:** Joint Health Systems Research Initiative Medical Research Council (MRC)/Department for International Development (DFID)/Wellcome Trust/Economic and Social Research Council (ESRC) (MR/P014844/1).

**Competing interests:** The authors have declared that no competing interests exist.

process made gains raising community 'voice' and initiating dialogue with the authorities, giving the voice 'teeth'. Achieving 'bite', however, requires longer-term engagement, formal and sustained connections to the system. Sustaining in highly fluid contexts and connecting to higher levels are likely to be challenging. Regular learning spaces can support development of collaborative

## Introduction

The social determinants of health paradigm cemented appreciation of fundamental causes of health inequalities [1]. While recognition of the need for integrated action on health within and beyond the health sector followed, working in silos and multi-sectoral *inaction* (inertia) remain the norm [2,3]. Similarly, despite acceptance of the social causes of health inequalities, marginalised community voices are seldom considered in practice settings [4,5]. The latter is despite long-recognition of community participation as integral to equitable health systems as part of a social justice approach [6,7]. Pervasive policy-implementation gaps reflect limited practical understandings of how to operationalise these important public health ideas.

In this paper, we report on a community-led, multisectoral, action-learning process in rural South Africa and make practical suggestions for integrating in practice settings as part of routine functions. The process focussed on alcohol and other drug (AOD) abuse, as a critical issue affecting rural communities experiencing serious and multidimensional hardship in the rural sub-district. Below, we provide an overview of health and health systems in South Africa in which we locate the research questions and objectives.

South Africa is an upper-middle income country with a population of 60 million, a third of which is rural. Life expectancy was 62 years in 2021 [8]. Since the end of the apartheid regime, a new political order set out to transform the health sector to one based on community-based primary health care (PHC), equitable provision, prevention and health promotion through radical nation-building reforms [9]. Health policy and strategy are described as 'near-ideal': progressive, inclusive, and pro-poor. In 2022, however, South Africa remains one of the most unequal countries in the world. Historical structural and racial injustices combine and converge with more current economic inequalities [10,11]. A generation after its emergence, HIV prevalence in black populations is 40–50 times that of white and HIV risks are eight times higher in adolescent females than males [12].

Entrenched health inequalities reflect and exacerbate wider social problems. The synergistic nature of HIV and substance abuse is well-documented [13–15] and alcohol and substance abuse is recognised as a serious, growing problem in the country [16]. Treatment demand for heroin-related problems is high in Mpumalanga, the rural province where the research was based [17]. Nationally, up to 15% of South Africans use drugs and a third of adults who consume alcohol report harmful use [18,19], with binge-drinking increasingly reported [20]. A significant increase in substance misuse among women has also been observed, although it is poorly understood [21].

There has been increased drug use among youth and adolescents, with severe increases in opioid-related disorders [22,23]. Over the past decade, *Nyaope* has emerged as a low-grade heroin smoked with marijuana, benzene, glue, and antiretrovirals (ARVs), used widely in rural areas. *Whoonga* is a distinct heroin-based street drug, made with household cleaning products, rat poison, and ARVs to which youth are seen as especially vulnerable [16]. There are limited resources allocated to the issue: low capacity, poor infrastructure, lack of information and poor intersectoral collaboration [24]. Targeted prevention and treatment is nevertheless seen as a priority and public health harm reduction approaches are urgently required [25].

The burden of AOD abuse exists in a public health system simultaneously dealing with the deeply unequal burdens of chronic comorbidities: 8.2 million known HIV positive in 2017, 2 million not on treatment (many with TB); declining treatment adherence; inadequate basic immunization; skyrocketing obesity; and mortality owing to injury and violence double the global average [8,26–28]. Serving the vast majority of the population, the public health system deals with this complex burden in the face of systemic underinvestment, human resource crises, corruption and deteriorating infrastructure [29].

Recognising deep-rooted social pathologies and a distinctly two-tier health system, a major district health systems revival is underway. Through Ministerial support for Universal Health Coverage (UHC), National Health Insurance (NHI) was introduced in 2012 with provincial guidelines for PHC Re-engineering [30]. In 2017, a policy framework and strategy for Ward-based Primary Healthcare Outreach Teams (WBPHCOTs) underscored commitments to *bringing services to people*: connecting vulnerable communities with health systems and devolving power to communities to create a more patient-focused and community-oriented system [31,32].

Within PHC Re-engineering, however, there is a familiar contradiction. Despite policy and strategy commitments to connect people and services, there are restricted operational spaces inclusive of marginalised community voices to understand and respond to local health priorities. In practice, frontline managers and providers are unable to engage with and respond to community priorities and local health planning seldom provides learning spaces. Participatory governance processes, such as clinic and health committees, do not function effectively and organisational 'compliance cultures' are seen as the norm, characterised by centrally-defined targets and outputs, with limited authority for local planning and management [4,5,33–35].

This paper is concerned with multisectoral action on the community-nominated local public health priority of AOD abuse. Despite progress in multisectoral working in COVID-19 responses, maternal, newborn and child health [3], non-communicable diseases (NCDs) [36], and HIV/AIDS [2], there remains limited understanding around *how* to achieve integrated, community-engaged action in practice: in relation to time, resources and power relations for system change inclusive of vulnerable communities, and in terms of contexts that constrain or enable impact [3].

In response, we sought to address the research questions: how can multisectoral action with communities be achieved and advanced?; what factors enable or disable it?; what institutional factors shape it?; and how can approaches be refined and expanded? The objectives were to: (a) support and enable multisectoral accountability for local public health problems through community-led action-learning; and (b) derive transferrable learning.

## Materials and methods

### Study setting

The study was progressed within the VAPAR (Verbal Autopsy with Participatory Action Research) programme (www.vapar.org). VAPAR combines verbal autopsy (VA), an established health surveillance tool to measure levels and causes of death where deaths are often uncertified [37], with participatory action research (PAR). PAR is a distinct methodology, which disrupts conventional subject-object distinctions in the health and social sciences, and whereby those most directly affected by problems adopt active roles as co-researchers and change agents, organising evidence for action and learning from action [38].

Combining these methods in an adaptive action-learning process, the programme addresses to two inter-connected problems: the lack of research evidence on the health needs of those socially excluded from access to health and information systems; and the lack of

uptake of evidence in the system by health service planners, managers, policy makers and providers at different levels.

VAPAR is based at the Agincourt Health and Socio-Demographic Surveillance System (HDSS) in Mpumalanga, a rural province of 4.7 million. Established in 1992, the HDSS covers 450 km$^2$ with a population of 120,000 encompassing settlements in former apartheid homelands typified by poverty and underdevelopment. The Agincourt HDSS is among Southern Africa's oldest and largest prospective population-based cohorts, and has supported the district health system over decades [39]. We worked in formal research partnership with planners, managers, and practitioners in the provincial Directorate for Maternal, Child, Women and Youth Health and Nutrition (MCWYH&N).

In the study area, there is a large population of orphaned youth experiencing severe impacts on wellbeing, familial support and school drop-out [40]. 16% of the population in the province is illiterate [41] and in 2019, unemployment in the district was 37% [42]. In the study area, villages vary in size from less than 5,000 to over 10,000. According to the policy prescript, wards are determined by village size, with WBPHCOTs catering for populations of approximately 7,600 people [30]. Depending on the ward, outreach teams consist of 6–10 community health workers (CHWs) and each CHW supports approximately 150–250 households. In practice, however, coverage is lower and uneven [43].

As described above, in post-apartheid South Africa, entrenched structural and systems challenges limit the potential of progressive health policy. This is seen across sectors, where the realisation of policy goals is undermined through uneven implementation [4,44]. Understanding and strengthening implementation is therefore urgently required. Moreover, in line with PHC re-engineering, community health brings attention to operational levels, local realities and needs, and feasible, local action; building inclusive, socially-accountable and resilient health systems [45,46]. While implementation can be improved through strengthening accountability, however, state-citizen relationships are not well understood [47–51].

In pilot work, we established that: (a) engaging community and health systems stakeholders in deliberative participatory processes promoted new forms of dialogue; and (b) including stakeholders beyond the health sector was necessary to address the social determinants of avoidable mortality [52]. On this basis, we set out to re-engage stakeholders representing communities and in the authorities in different levels and sections to expand the pilot process. The process sought to support and enable mutual accountability as a means to address, and ultimately close, policy-implementation gaps.

We designed an adaptive action-learning process as follows: *Step 1*—engage community stakeholders to identify and generate evidence on local health priorities; *Step 2*—initiate dialogue between communities and the authorities to build mutual understanding, analyse data and plan remedial action; and *Step 3*—collectively act on data generated, and collectively reflect on and refine the process. The cycles were designed to be iterative and adaptive to the local context. Our theory of change was that repeated cycles of action and reflection would enable relationships and collective capacities, supporting improvements in service delivery and health outcomes and with potential for sustainability and wider uptake (Fig 1) [53]. In this paper, we report on the first cycle.

## Data collection

**Step 1: 'Engage/Observe'.**   To maintain prior relationships, we engaged community stakeholders involved in the pilot in villages across the three clinical catchment areas in the Agincourt HDSS. Three villages were selected based on demographic variation, socioeconomic status, and accessibility to health services. Within each village, participants were selected to

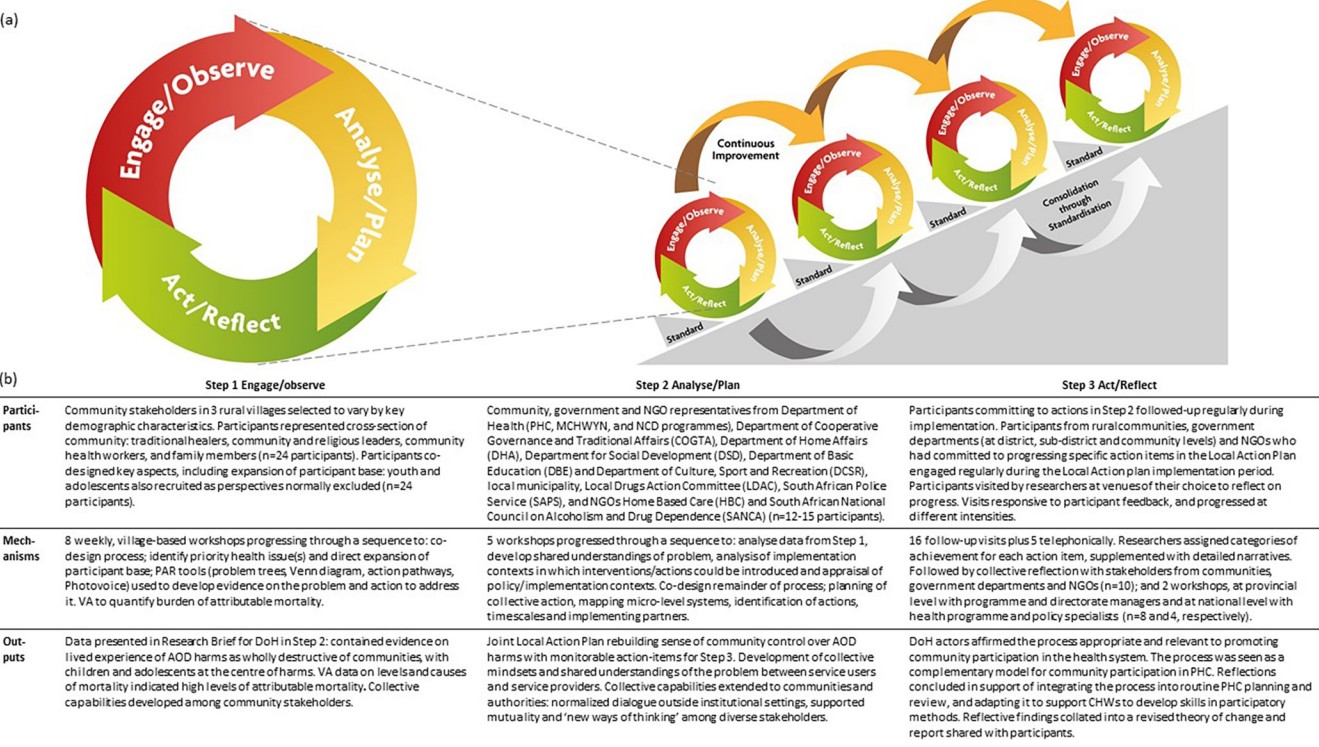

| | Step 1 Engage/observe | Step 2 Analyse/Plan | Step 3 Act/Reflect |
|---|---|---|---|
| Partici-pants | Community stakeholders in 3 rural villages selected to vary by key demographic characteristics. Participants represented cross-section of community: traditional healers, community and religious leaders, community health workers, and family members (n=24 participants). Participants co-designed key aspects, including expansion of participant base: youth and adolescents also recruited as perspectives normally excluded (n=24 participants). | Community, government and NGO representatives from Department of Health (PHC, MCHWYN, and NCD programmes), Department of Cooperative Governance and Traditional Affairs (COGTA), Department of Home Affairs (DHA), Department for Social Development (DSD), Department of Basic Education (DBE) and Department of Culture, Sport and Recreation (DCSR), local municipality, Local Drugs Action Committee (LDAC), South African Police Service (SAPS), and NGOs Home Based Care (HBC) and South African National Council on Alcoholism and Drug Dependence (SANCA) (n=12-15 participants). | Participants committing to actions in Step 2 followed-up regularly during implementation. Participants from rural communities, government departments (at district, sub-district and community levels) and NGOs who had committed to progressing specific action items in the Local Action Plan engaged regularly during the Local Action plan implementation period. Participants visited by researchers at venues of their choice to reflect on progress. Visits responsive to participant feedback, and progressed at different intensities. |
| Mech-anisms | 8 weekly, village-based workshops progressing through a sequence to: co-design process; identify priority health issue(s) and direct expansion of participant base; PAR tools (problem trees, Venn diagram, action pathways, Photovoice) used to develop evidence on the problem and action to address it. VA to quantify burden of attributable mortality. | 5 workshops progressed through a sequence to: analyse data from Step 1, develop shared understandings of problem, analysis of implementation contexts in which interventions/actions could be introduced and appraisal of policy/implementation contexts. Co-design remainder of process; planning of collective action, mapping micro-level systems, identification of actions, timescales and implementing partners. | 16 follow-up visits plus 5 telephonically. Researchers assigned categories of achievement for each action item, supplemented with detailed narratives. Followed by collective reflection with stakeholders from communities, government departments and NGOs (n=10); and 2 workshops, at provincial level with programme and directorate managers and at national level with health programme and policy specialists (n=8 and 4, respectively). |
| Out-puts | Data presented in Research Brief for DoH in Step 2: contained evidence on lived experience of AOD harms as wholly destructive of communities, with children and adolescents at the centre of harms. VA data on levels and causes of mortality indicated high levels of attributable mortality. Collective capabilities developed among community stakeholders. | Joint Local Action Plan rebuilding sense of community control over AOD harms with monitorable action-items for Step 3. Development of collective mindsets and shared understandings of the problem between service users and service providers. Collective capabilities extended to communities and authorities: normalized dialogue outside institutional settings, supported mutuality and 'new ways of thinking' among diverse stakeholders. | DoH actors affirmed the process appropriate and relevant to promoting community participation in the health system. The process was seen as a complementary model for community participation in PHC. Reflections concluded in support of integrating the process into routine PHC planning and review, and adapting it to support CHWs to develop skills in participatory methods. Reflective findings collated into a revised theory of change and report shared with participants. |

**Fig 1.** (a) VAPAR schema: adaptive action-learning cycles to coproduce and embed evidence in practice to improve care and outcomes; (b) Action-learning cycle detailing each stage of the process. The schematic diagram in Fig 1 was produced by the research team.

represent a cross-section of the community including traditional healers, community and religious leaders, family members and community health workers (CHWs) (n = 24) (reported elsewhere [54]). Of the 24 former participants approached, 13 agreed to be involved and 11 additional individuals were recruited (see *Ethical considerations*, below).

We held an introductory workshop in which participants were invited to co-design key aspects of the process: nominating priority health concerns and directing expansion of the participant base to include otherwise excluded perspectives. Alcohol and other drug abuse (AOD) was nominated as a priority health concern locally, and youth and adolescents were identified as a specific group directly affected by, and whose voices were excluded from attention to, the problem. Community stakeholders then connected the researchers to 24 additional participants, known in the community to represent youth and adolescent perspectives. These individuals were approached and recruited by researchers (see *Ethical considerations*, below).

A series of three workshops was then held in each village, with a further four bringing participants from all villages together (August-September 2017). PAR methods were employed to: (a) systematise subjective experiences into shared accounts of the problem; (b) build shared accounts reflecting consensus on cause-and-effect relationships; (c) explore locally-acceptable actions to address the issues identified; and (d) articulate 'action agendas', i.e. overall goal(s) and stepwise actions and actors to achieve these [38]. Visual evidence was also collected and appraised [55]. Photographs collected by participants to convey lived experience were presented and discussed in the workshops.

The workshops were also designed to support participants to assume ownership and control over the process. Participants directed practical aspects such as times and venues for workshops, shared the facilitation and recording of proceedings, and we regularly discussed

broader participatory principles of representation, voice, collective action, power-sharing. Participants cross-verified analyses and reflected on the process and its further development throughout, and in a dedicated session at the end of the sequence.

Statistics on mortality attributable to AOD abuse were also compiled by the research team using VA data from the Agincourt HDSS to provide additional evidence on the scale of the burden of mortality owing to community-nominated priorities. Data were acquired on all deaths 2014–15 and processed using InterVA-4 [56]. The VA and PAR data were compiled into a 'DoH Research Brief' for the next step (S1 Text).

**Step 2: 'Analyse/Plan'.** To initiate dialogue between communities and the authorities, a series of 'analysis' workshops was then held (January-April 2018), to present and appraise the VA and PAR data generated in the prior step. The first workshop was held in DoH offices in the City of Mbombela with 13 officials from provincial and district programmes. We reviewed data from Step 1 and codesigned the remainder of the process, agreeing on multi-sectoral representation. Two further analysis workshops convened 12 and 15 participants respectively: community stakeholders, representatives from provincial and district DoH and from departments such as education, social development, and culture, sports, and recreation. Following presentation of the VA and PAR data via the research brief and by research and community partners, we used group model building to develop a collective definition of the focus and boundaries of the problem, formulate problem statements, consider potential strategies to address the issues, and appraised policy contexts [57,58].

Thereafter, two 'planning' workshops were held with representatives from communities and authorities to appraise and organise collective action addressing the issues identified. These workshops were located in the Agincourt HDSS field offices to bring the focus to implementation and the rural sub-district (September-December 2018). Fifteen (15) participants were engaged, a combination of those already involved and local implementing partners: sub-district DoH, local municipality and local non-governmental organisations (NGOs). In the 'planning' workshops, we developed the prior 'analysis': to understand and operationalise multisectoral responses. Through facilitated group work, we mapped the micro-systems in which interventions could be introduced and identified potential entry and leverage points through strategic and operational analyses. We then developed and appraised action in terms of affordability, acceptability, regulatory impact, fit with other policies, implement-ability, timescales, and health and social impacts. This culminated in the development of a joint Local Action Plan that all participants had active roles in and made formal commitments to, and that was circulated in a written report.

**Step 3: 'Act/Reflect'.** Participants who committed to specific action-items in the Local Action Plan were followed up over a subsequent 4-month implementation period (January-April 2019). Participants were visited by researchers at venues of their choice. Visits were responsive to participant feedback and progressed at different intensities. A structured tool was used to capture mechanisms through which change did or did not occur (S2 Text). The research team collated data on implementation and reflected weekly on the extents, benefits, and risks of action, and on the process and its acceptability. A total of 16 follow-up visits with participants from rural communities, government departments at district, sub-district and community levels and NGOs were completed with an additional five telephonically. At the end of the monitoring period, the researchers assigned categories of achievement for each action item, supplemented with detailed narratives.

The cycle culminated in a group reflection. In April-May 2019, we interviewed 10 participants from communities, government departments and NGOs, and held two workshops with health systems stakeholders (n = 8 and 4 participants, respectively). We sought perspectives on whether and how impacts had been achieved; acceptability and utility of the process; levels and

mechanisms for integration into the health system; and future linkages. The researchers developed a revised theory of change in a report shared with participants.

## Analysis

This paper presents an analysis of whether and how the process enabled and supported mutual accountability for local health concerns among community and health systems stakeholders. The main focus was the idea that health inequalities are fundamentally driven by structural factors over which individuals have little or no control. From this perspective, health inequalities are social issues with social, as opposed to individual, causes and can only be solved with collective action [59,60]. The research was therefore located within structure/agency theoretical debates and the reframing of wholly individualistic explanations.

We used *strategic social accountability* developed by Fox focussed on strengthening citizen voice *in parallel* with supporting state capacity to respond [48]. To deepen the focus on processes and interfaces between actors and institutions, we also used structuration theory [61]. Structuration resolves the fundamental structure-agency conflict in the social sciences by recursively linking them through social processes. Structuration thus attends to *how* individual agents are influenced by social structures and, how similarly and simultaneously, social structures are shaped by the actions of individual agents. Structuration theory has been used extensively in organisational and management studies and its relevance in public health is recognised [62–65].

The main data sources were narrative and visual data, workshop reports, reflective journals, and formal and informal participant feedback supplemented with presentations, registers, study protocols and individual and team reflections. Researchers familiarised with the data, immersed in data sources, performed an initial organisation according to analytical (deductive) categories: forms and dynamics of the process, actor interactions and interfaces, impacts and degrees of impacts, enabling and constraining contextual factors, as well as to emergent (inductive) themes. We reviewed the analysis regularly, creating and assigning themes, and considering relationships within and between them to build a reflexive account related to the theory of change. The analysis was partly progressed through group reflections and analysis, and partly by organising and analysing material using the data management package NVIVO.

## Ethical considerations

Rooted in participatory theory and method, ethical considerations related to sharing of power during the process, and the continual reflection on and development of principles of representation, social justice, lived experience and collective action. These were embedded into the overall design, which explicitly sought to shift power towards those with little or none regarding the priority issue identified. Similarly, every workshop and engagement re-visited principles of power, control, action, and representation. Otherwise, all participants were provided with written information on the research with contact details for the research team and given minimum 72 hours to consider and ask questions. Written consent was sought from all participants in which we assured anonymity and that participants were free to leave the process at any time and for any reason. Participants were provided with refreshments, transport costs and were reimbursed for time spent participating in workshops: 300ZAR per participant (Step 1), 250ZAR and 200ZAR per workshop (Steps 2 and 3 respectively). All workshops were held in agreed locations, at convenient dates and times and in local languages where appropriate and as directed by participants. For the visual PAR component, participants received basic training in photography, and on how to obtain permissions from subjects of photographs. Finally, institutional boards at the University of [anonymous] (M1704155; M171050) and

University of [anonymous] (CERB/2017/4/1457; CERB/2017/9/1518) reviewed and approved study protocols and the provincial health authority gave permission for the research (MP_201712_003). Data were stored on secure servers at the Universities of the [anonymous] and [anonymous].

## Results

The results are presented by emergent themes and, within this, by analytical constructs drawn from the frameworks and theories adopted. These were: process forms and dynamics; actor interactions and interfaces; impacts and degrees of impacts; and enabling and constraining contextual factors. Where possible, the analysis is illustrated with verbatim quotes from participants. This is followed by an account of transferrable learning.

### Regular community engagement in rural villages built rich and vivid community intelligence

In Step 1, community stakeholders developed sophisticated, multi-level accounts of AOD abuse as an entrenched social problem. Distinct dynamics were observed and expressed in rich and vivid discussions and accounts of AOD abuse, depicting a reciprocal, self-sustaining issue with destructive effects on families and communities, and with children and adolescents placed at the centre of exposure to risk and lifelong consequences (Step 1 on AOD abuse is reported in detail elsewhere [66]). Researchers supplemented this with VA data, in which approximately 30% of all deaths were attributed, at least in part, to AOD abuse.

> . . .alcohol is destroying our communities and families. . .if we continue like this our children will not have a better future; our level of education will remain poor forever [Community stakeholder, Step 1]

> It makes me happy to see that two villages have chosen to talk about drugs and alcohol which shows us that it doesn't affect certain people only but all villages. All we want is to see our children recovering, we want to see our children living a normal life because if they continue using drugs and alcohol their future is destroyed, which also destroys the future of the whole community [Community stakeholder, Step 1]

Interactions changed markedly over the course of the village-based workshops in Step 1. Initially, significant dissatisfaction and mistrust were expressed by community stakeholders towards the authorities. Willingness to collectively advance solutions emerged, however, through repeated interactions that were action-oriented and focussed on ideas of mutual accountability and collective action. The PAR tools and principles provided a framework for this: moving from identifying and problematising the issue in terms of subjective perspectives, through to building collective accounts, mindsets, and shared agendas to address the issues identified.

Impacts in Step 1 were observed in terms of familiarity, ownership, and control, which built over the course of the workshops. This was supported and enabled by prioritising prior relationships, co-designing the process, regular dialogue, and engagement, and locating workshops in accessible areas and at reasonable times. New participants brought a needed perspective, but also challenges, reflecting aspects of wider social contexts. Some youth participants were initially sceptical about the potential for change and often disruptive during the workshops.

*We have learned some of the things, but let's face the main issue—this process won't change anything, there's a lot of things that won't change [Community stakeholder, Step 1]*

In response, we reinforced principles of respectful engagement and took time to manage expectations. We discussed and agreed that, while the process sought to build appreciative understandings of what works or does not and why within a mutually supportive 'learning platform', change could not be guaranteed. With sensitive facilitation that was responsive, compassionate, and consistent more constructive and collective attitudes emerged. The resulting VA and PAR data and evidence were clear, actionable, and seen as legitimate sources of evidence by participants. A research brief containing VA and PAR data was produced for the subsequent workshops with the authorities (S1 Text).

*. . .this research will improve our community and we gain knowledge, we learned about caring for ourselves and to work together with other people [Community stakeholder, Step 1]*

*. . .we know the statistics of people using drugs and alcohol it is easier to work with something that you are sure of [Community stakeholder, Step 1]*

## Time, space, and a sensitive, inclusive approach supported multisectoral dialogue

The VA and PAR data and evidence were then used as the basis of dialogue between rural communities and the authorities. Here, contextual, and organisational influences were again evident in the interactions and interfaces. The 'analysis' workshops were initially located in provincial DoH offices, which we quickly realised may have restricted participation, especially among community representatives, owing to these being far away, highly professional spaces. Nevertheless, key dynamics were observed. Government participants were responsive to the process, verifying and remarking upon the consistency of their analysis with that of the communities'. Government stakeholders were also enthusiastic about communities reporting on their situations and taking active roles in health action.

*I have benefitted a lot from the approach of this project. The workshops displayed the importance of real participatory approach in research not just the theoretical approach [DoH stakeholder, Step 2]*

Reflections of mutuality deepened when operational contexts in which interventions could be introduced were examined. DoH stakeholders highlighted the need for multisectoral approaches to tackle AOD abuse and an urgent need to regulate taverns. CHWs were identified as vital support for rural communities grappling with AOD abuse. In open and frank discussions, multiple obstacles to cross-sectoral integration were described with candid accounts provided in the mapping sessions. Participants found that departments shared policies addressing similar problems with similar objectives and aims. However, the impracticality of some policies, resource shortages, donor-driven priorities 'crowding out' local needs and priorities, and frequent policy revisions were identified as obstacles to implementation. In response, participants indicated that multisectoral collaboration among various government departments, communities and researchers is needed to improve health through development of relevant programs and interventions that are tailored to the needs of communities. Participants also raised concerns over lack of attention to how organisational cultures impact implementation, and outcomes (S1 and S2 Tables).

Addressing potential constraints on engagement and dialogue in overly 'professional' spaces, the location of the subsequent 'planning' workshops in the Agincourt HDSS field offices helped shift the focus to rural implementation contexts. In these workshops, we jointly developed a time-bound, monitorable Local Action Plan that was owned by and shared among participants. Community stakeholders committed to mapping AOD abuse hotspots and strengthening collaboration between traditional leaders and police to regulate taverns. District-level DoH stakeholders committed to encouraging professional nurses to provide information and support in schools. Other governmental representatives at operational levels committed to improving application of substance abuse legislation in alcohol outlets, disseminating information on social support and mobilising resources for a community-based rehabilitation facility (S3 Table).

In terms of impacts, community and government stakeholders reported appreciation of neutral 'safe spaces' outside institutional environments to connect with and understand other stakeholders and remarked on how this supported 'new ways of thinking'. Multisectoral engagement inclusive of communities was challenging, however, both logistically and in the discussions and reflected contextual constraints. Creating spaces outside institutional environments was well-received but required time to build and maintain relationships and sensitive facilitation to support inclusive dialogue. This posed challenges for already over-burdened practitioners and service planners. Again, sensitive, and respectful engagement supported repeated engagements, which in turn provided the time and space to reinforce principles of cooperative learning, normalised dialogue, and established confidence with the process among diverse participants.

*. . . [the workshops] created an awareness and gave more insight on the gaps the Departments may be faced with when delivering their mandate in the respective disadvantaged communities. More still needs to be done and Departments should play a major role in implementing activities that can benefit these communities [Government stakeholder, Step 2]*

*. . .the workshops were all effective considering that they brought in stakeholders from various departments and organizations [Government stakeholder, Step 2]*

*The greatest gain was insight into the functioning of other sectors and Departments [DoH stakeholder, Step 2]*

## Appreciative monitoring supported collective action

Over the implementation period, there was mixed progress with the action plan. Of the six action items, one was achieved, four were partially achieved and one was not progressed (Table 1). The action-item that was achieved was the mapping of substance abuse hotspots. This was led by the youth representative of the community actors supported by the researchers, among whom there was an arguably higher 'stake' in the process, coupled with a short-term, relatively achievable outcome.

Of the four partially completed action items, two involved actors at national level (drug laws and compliance in taverns), one depended on departmental funding (social support services dissemination), and one was longer-term in nature, and subject to shifting priorities and spending (community rehabilitation centre). Collective reflection on these helped achieve deeper, shared understandings of how, where, with whom and to what extent action was feasible. Elsewhere, some activities were to an extent already planned or in progress. While there may have been merit in bringing them together in a multisectoral plan, progress could not be

**Table 1. Progress towards the Local Action Plan.**

| Action item | Outcome | Implementation mechanisms |
|---|---|---|
| Community stakeholders (leads) | | |
| 1. Identify AOD abuse hotspots to focus action and aid departments planning interventions | Achieved | A home-based carer (CHW category) working outside the study area was nominated to lead this action as a community stakeholder representative. The researchers found it hard to engage with her during the monitoring period, however, and through the home-based care organisation connected with a home-based carer working within the study area who was a community stakeholder. Together with other community stakeholder representatives, they validated a list of AOD abuse hot spots (shebeens, taverns) obtained from a study in the Agincourt HDSS, and added hotspots that were not on the list. While community leaders were not convened due to lack of resources and other commitments, the action was completed by other means, with verification of the hotspots by community stakeholders. |
| 2. Strengthen collaboration between traditional leaders and police service to regulate taverns | Some progress | Follow-up visits highlighted various and variable connections between traditional leaders and law enforcement. The action is happening in the sub-district in which the Agincourt HDSS is based, but not in the study area. The Local Drugs Action Committee (LDAC) (a body consisting of government, SAPS, and NGO partners) works with traditional leaders encouraging community members (especially men) to attend LDAC information sessions. To further understand the connection between traditional leaders and law enforcement, we linked to Community Policing Forums (CPFs) in the study area to find out how they work with police and report cases of AOD abuse. Through this, we established that CPFs in the study area do not know about LDAC. We also engaged with MER (Mpumalanga Economic Regulator previously liquor board) that works with traditional healers, SAPS and municipalities when issuing licences. Traditional leaders endorse the application while SAPS checks for criminal records on all applicants. |
| Department of Health (leads): district, sub-district and community levels | | |
| 3. Encourage nurses to adopt schools (primary and secondary) | No progress | Stakeholders leading the process reported multiple and competing priorities during implementation follow up, which precluded action. In response, the researchers met with DBE in March 2018 to find out more about school health services, which revealed a gap between services mandated and delivered, as well as providing insights into where NGOs such as SANCA are working with the authorities to deliver health promotion in schools. DBE advised that while DoH policy supports nurses in schools, in practice, nurses only provide services when required. Otherwise, SANCA provides health promotion in schools, albeit variously, and in partnership with DoH, SAPS and DSD. |
| Other departments and agencies (leads): district, sub-district and community levels and NGOs | | |

*(Continued)*

**Table 1.** (Continued)

| Action item | Outcome | Implementation mechanisms |
|---|---|---|
| 4. Align Department of Justice (DoJ) and SAPS on application of substance abuse legislation in liquor outlets | Some progress | This is a long-term outcome. The policy is under review at national level. However, we engaged with MER, whose main duties are compliance and licensing, to understand the process of licensing and issues of compliance. MER runs programmes with SAPS, DBE, HSRC (Human Science Research Council), municipality and traditional leaders. MER also work with SANCA and other organisations on public awareness in schools and communities. MER have inspectors in the sub-district on compliance and responsible trading. The researchers established a relationship with MER and they are willing to attend our workshops when invited in the next action-learning cycle. |
| 5. Disseminate information on AOD abuse activities | Some progress | DSD had in their annual plan to host the 'Blitz' event however it was not done due to lack of funds. At the time of reporting, the DSD are awaiting the subsequent annual budget and seek to highlight the need for role clarification and dissemination of information on provisions for social support identified through the process in department planning in the subsequent period. |
| 6. Mobilise resources for community-based rehabilitation centre | Some progress | There were no applications for treatment centres received by DSD at province level by the time of reporting. However, the district level is assisting and supporting willing organisations to apply, none of whom are in the study area. A potential role for the researchers was discussed during implementation follow up to mediate and support the applicants to keep pace/motivation through a mid-term process. |

CPF: Community Police Forums; DBE: Department of Basic Education; DoJ: Department of Justice; DSD: Department for Social Development; HSRC: Human Science Research Council; LDAC: Local Drugs Action Committee; MER: Mpumalanga Economic Regulator; SANCA: South African National Council on Alcoholism and Drug Dependence; SAPS: South African Police Service.

solely attributed to the process and was often outside the control of those who had committed to the action.

Reflecting constraining contextual factors, one action-item was not achieved: health sector support for schools. This activity had been displaced owing to multiple, competing priorities among nurses at PHC level. Despite policy support for integrated school health services, overall budget restrictions have led to limited staff. Combined with the commitment not being assigned as a key performance area, it amounted to encouraging nurses to adopt schools as part of their social responsibility, which was not feasible in practice. Coordinating with governmental priorities was identified in response, as critical to support action and impact, which helped to build a more strategic view which ultimately supported a deeper degree of impact (see below).

In terms of actor interactions and interfaces, during implementation of the Local Action Plan, researchers quickly became aware of punitive implications of 'follow-up' monitoring particularly among DoH stakeholders. This was especially the case where action was challenging and did not progress as planned. In response, we reinforced that the process was appreciative: it was not about fault-finding but focussed on building relationships and shared understandings about needs, priorities and what supports or constrains action.

Process forms and dynamics expanded during implementation, as participants connected researchers to additional actors. Researchers developed new connections with the Department for Basic Education, who confirmed gaps between mandated and delivered services and provided insights into where NGOs were working with the authorities to deliver health promotion in schools. We also connected with the Mpumalanga Economic Regulator (MER), sub-regional managers of the local municipality, and the community policing forum (CPF): a platform where community organisations, provincial government, local government, traditional authority and parastatals and the police meet to discuss local crime prevention initiatives.

## Collective reflection, adaption and coordination with government enabled uptake

In the final, reflective element, the importance of aligning to government priorities was highlighted. DoH actors affirmed finding the process appropriate and relevant to promoting community participation in the health system. Reflecting on limited and variable functionality in formal community engagement processes, the process was seen as a complementary model for community participation in PHC. In terms of impacts, these discussions and reflections concluded in support of integrating the process into routine PHC planning and review and adapting it to support CHWs to develop skills in participatory methods (Table 2).

Otherwise in term of impacts, community stakeholders expressed more constructive attitudes about the authorities, together with beliefs about self-efficacy, and reported building strategic, analytical, and public-speaking skills and confidence as a result of the process (Figs 2 and 3). Government stakeholders gave positive feedback reporting their enthusiasm about opportunities to meet, learn about and engage with other departments, which was reported to enable linkage and collaboration. They responded positively to the methods employed during the workshops. Government stakeholders found it encouraging and intriguing to hear community voices directly, enjoyed engaging with the community and seeing communities adopt roles as active change agents. Government stakeholders also appreciated the combination of statistical

**Table 2. Collective recommendations on integration into routine health systems processes.**

| Stakeholders | Recommendation |
|---|---|
| Government, NGO and community stakeholders | • Include local municipal managers during all stages of prospective action-learning cycles<br>• Convene stakeholders at the end of each VAPAR cycle for collective reflection and learning |
| Provincial DoH stakeholders | • PHC clinic operational managers and CHWs to be included at all stages of the next action-learning cycle, with a focus on skills exchange<br>• VAPAR representatives to participate in routine district and sub-district planning and reporting processes, including development of the district health plan and quarterly performance review<br>• Alignment/integration of VAPAR programme into existing health structures at critical levels of engagement; primarily at household/community (CHW/WBPHCOT) and sub-district (clinic operational managers, PHC supervisors) level<br>• Focus on community participation and contemporary priorities—support strengthening the management model of PHC facility manager, and consider other programmatic priorities such as adolescent and mental health |
| National DoH stakeholders | • Refinement of VA with regards to place of death/circumstances of mortality<br>• Continued engagement at national level to report on progress and inform future development/application and feeding up into national learning |

Source: [67].

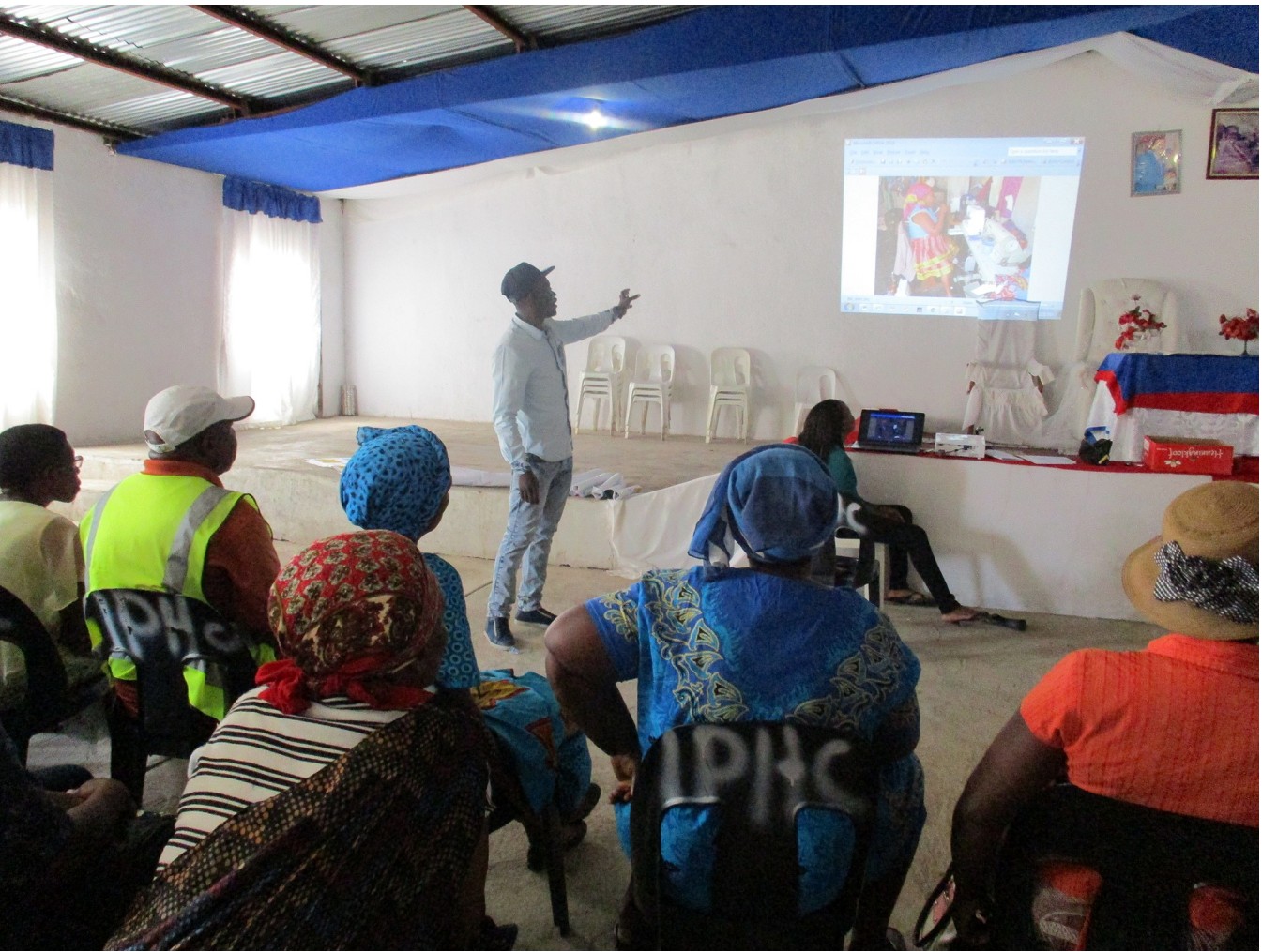

**Fig 2. Community stakeholder presenting and appraising visual evidence.** Image owned by the VAPAR programme and reproduced with permission in this paper under a creative commons license. Permissions have been secured from participants for the reproduction of all images taken during the course of the research.

and qualitative (including visual) data combining 'hard' data on burden of disease, with evidence on lived experience (Fig 4).

In terms of dynamics and interfaces, and while the action plan was only partially achieved, the process overall was seen as valuable: engaging a diverse and otherwise disconnected set of stakeholders in 'safe spaces' where difficult conversations could occur, and where shared awareness of local priorities could be built. Overall, there was good participation, high levels of workshop attendance and engagement in follow-up discussions. Participants reported seeing the platform as able to foster learning opportunities and new ways of thinking:

*The workshops reconfirmed that community participation is key to planning and improving service delivery [Government stakeholder, reflective workshop]*

*There have been a lot of service delivery protests in communities, but they did not accomplish much; everyone realized that it is time to shift our ways of thinking and initiate dialogue, unite and collaborate and create sustainable partnerships to solve community problems [Community stakeholder, reflective workshop]*

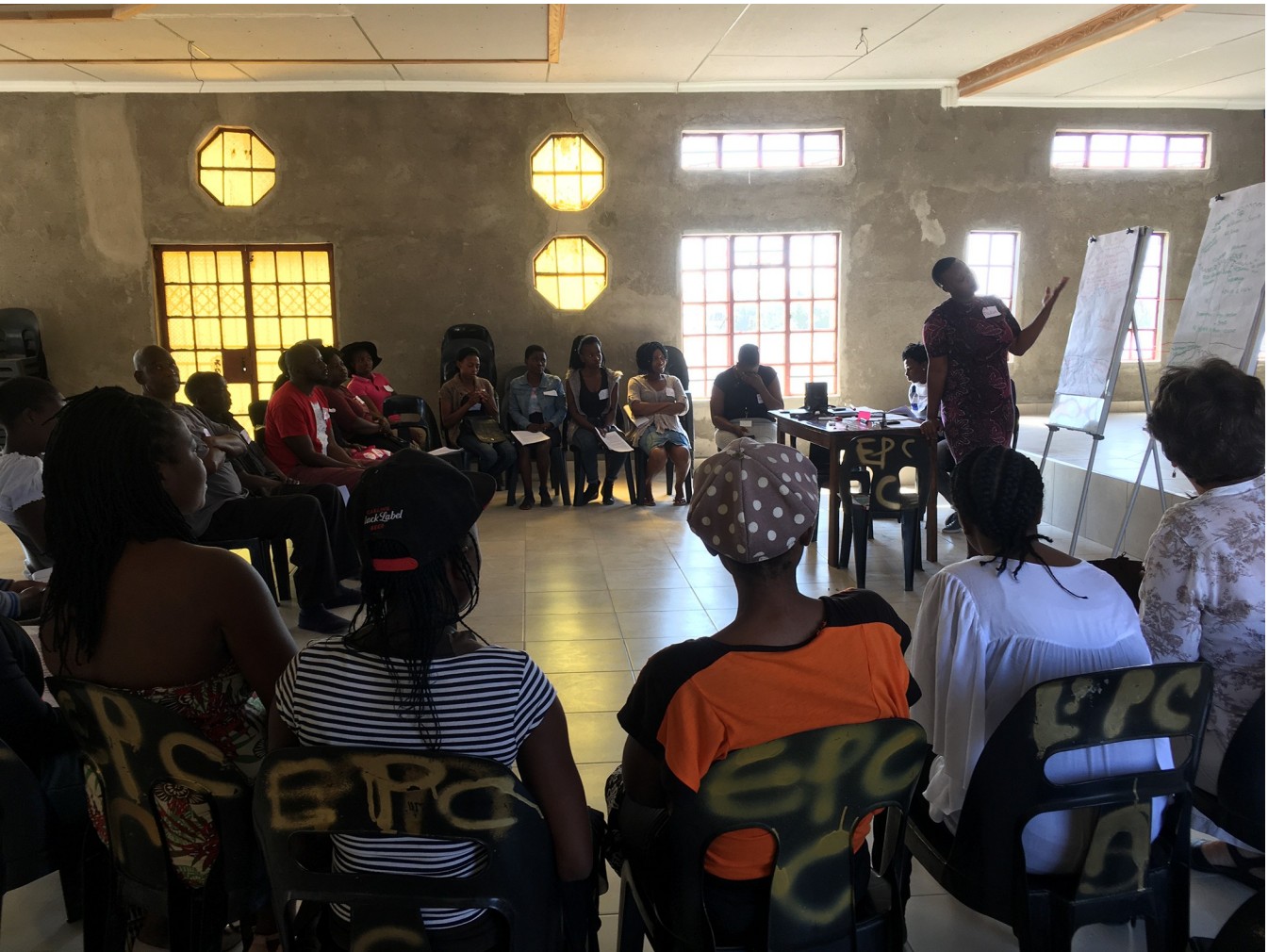

**Fig 3. Community stakeholder facilitating a group discussion on AOD abuse as a priority local health concern.** Image owned by the VAPAR programme and reproduced with permission in this paper under a creative commons license. Permissions have been secured from participants for the reproduction of all images taken during the course of the research.

Revisiting assumptions about how, where, for whom and to what extent change occurs, we revised the theory of change with explicit attention to integrating with routine systems, acting on data and evidence as an input to services, learning, skills exchange, shared priorities, improved awareness of local health priorities, sustaining action, and roles, and on how rural PHC and research contexts present challenges and opportunities (presented elsewhere [67]). The collective reflection was reported acknowledging all participants in a written report.

## Discussion

### Transferrable learning

Our initial theory of change was that a series of adaptive action-learning cycles could support development of capacities and relationships to enable recognition and uptake in the health system [53]. Inclusivity was a key *input*. With a focus on marginalised voices, we initially worked with community stakeholders developing collective capabilities to raise and frame local health concerns. Stakeholders from the authorities were then engaged to build mutual

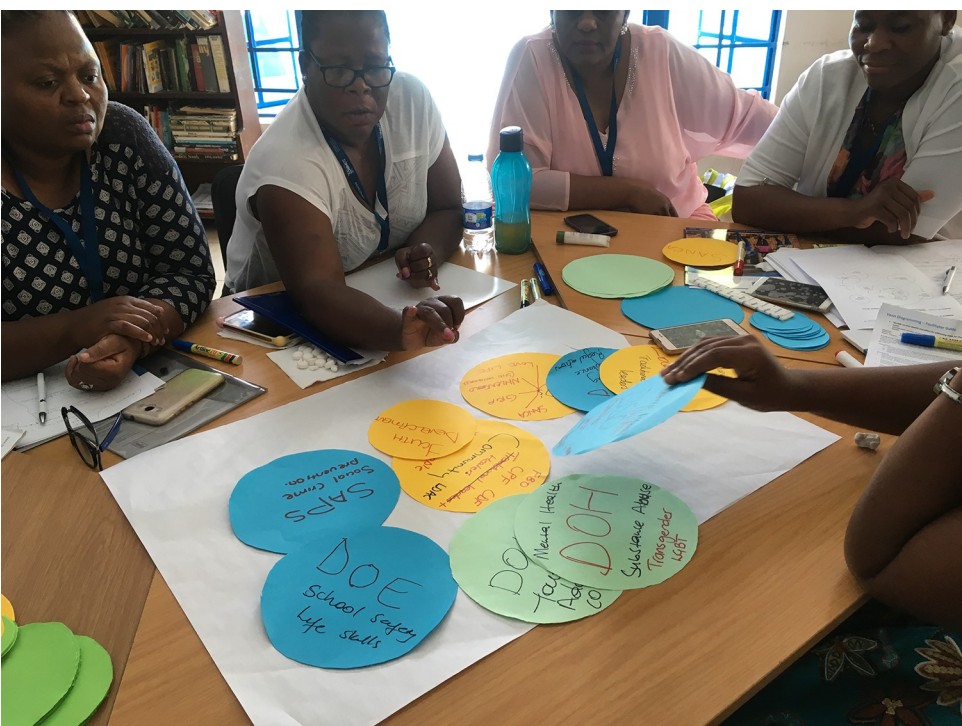

**Fig 4. Government, non-government and community stakeholders map key actors and institutional contexts to address issues identified.** Image owned by the VAPAR programme and reproduced with permission in this paper under a creative commons license. Permissions have been secured from participants for the reproduction of all images taken during the course of the research.

understandings of the issues, appraise policy and systems responses, and identify how, where and with whom collective local action could address the issues identified.

Throughout, inclusive representation and the criteria through which representatives were appointed were defined and negotiated. Continuous efforts were made to nurture 'voice', power, and agency among those with little or none. Initially, males and females representing rural communities had generally similar ethnicities, racial and social status. Representation was further expanded by participants themselves to draw in excluded perspectives. This supported development of collective capabilities and voice, which provided a necessary foundation for engaging with the authorities. Subsequent engagement with service providers at strategic and operational levels (province, district, sub-district and local) introduced a range of power asymmetries related to various forms of formal and informal status and privilege. We reconfigured and adapted processes and tools to accommodate everyone regardless of status, embracing diversity and enabling mutuality.

The Agincourt HDSS also brought additional 'hard' data to bear on the mortality burden related to community-nominated priorities. Such data are essential for public health planning, but their availability and quality are often lacking, especially on community deaths. Credible, actionable information was a further input that supported multisectoral engagement with the authorities. As stable public health observatories, HDSSs occupy strategically important positions to broker data-driven decision-making between communities and authorities.

Regular engagement was a critical *mechanism* that helped develop collaborative mindsets. Repeated 'safe space' engagements first with community representatives, then with the authorities, helped build dialogue, relationships, and shared understandings. The Agincourt HDSS

was seen as a neutral space to bring together service users and providers, between whom there were clear divisions and tensions. With consistent and sensitive facilitation, it was possible to develop mutual accountability and reflect on the fit and functionality of the process, adapting and reconfiguring as necessary. Recognition of and space for shared facilitation was important to bridge cultural and power differences. While the researchers developed enabling environments and brokered interfaces, these practices can and should be diffused through distributed, dynamic processes responsive to local contexts.

The Local Action Plan was a crucial *output* that ratified shared understandings and commitments to work together on common problems. While progress was mixed, and monitoring was identified as potentially punitive among some participants, regular time and space for shared reflection supported principles of mutual accountability and action. Collectively reflecting on and adapting the process was pivotal in supporting shared ownership and control, and enabled uptake into formal service planning and management. The codesigned adaptions will structure future cycles, further testing these inputs, mechanisms, and outputs, with deeper reflection on contextual conditions and whether and how change can be achieved and sustained (Table 3).

While suggestive of impact, the results should be interpreted with caution in terms of the relative strengths and limitations of the process. Firstly, while designed to be jointly owned, researchers controlled several aspects: overall design, resources, monitoring implementation of the action plan, and preparing outputs such as manuscripts and other briefing materials. Nevertheless, and while the action plan was only partly achieved, the *process* was clearly acceptable, and of practical value and benefit to participants. Reflecting on *instrumental and intrinsic* value in accountability processes, Joshi highlights the importance of how, and by whom, success is defined in social accountability processes [49]:"... *if a social accountability intervention fails in improving services but scores highly on empowerment of citizens, do we consider it a failure or a success, given that the intervention has changed the long-term prospects for accountability by changing the starting point for the next intervention? Whose definition of outcomes count?"*

In terms of the researchers' positionality, the team was primarily South African, based at Agincourt HDSS or Mpumalanga Department of Health and in the UK affiliated to Agincourt. We are a long-standing collaboration with a shared commitment to distributed and evidence-informed decision-making in rural PHC. Within the team, we considered practices, such as who controls funding, collects and analyses data, who publishes, and whose perspectives are prioritised [68]. This was informed and supported by the Agincourt HDSS, a centre grounded in commitments to rural communities over decades [69,70]. As noted above, HDSSs occupy strategically important positions and can play important roles connecting service users and providers and providing robust data. Further embedding in the HDSS will support development of transferrable processes to align research to national and sub-national priorities, and support accountability of researchers to local contexts.

In future iterations, more explicit recognition that design choices overall are underpinned by ideological and epistemological positions is needed, as is attention to categories of power and politics in evidence and action on health inequalities [50,51,71–73]. Specifically, testing and revising assumptions about how, where, for whom and to what extent change occurs should be driven in future cycles *by the collective*. As described above, limitations of social accountability relate to sustaining collective action among multiple stakeholders to achieve and understand change in long-term processes [74]. A learning approach that is emergent, built in and for local contexts, inclusive of collective articulation of theories of change, attention to power and sustained action will support relevant and sustainable processes enabling mutual accountability.

**Table 3. Transferable learning on community-led multisectoral action learning.**

| Theory of Change | Analytical domain | Transferable learning |
|---|---|---|
| Inputs | Actors (agents) | • Inclusivity is fundamental, representation should be continually negotiated, focussed on those most marginalised, excluded and hard to reach;<br>• Communities generating information on their own situations confers collective efficacy, action and generates credible, actionable information;<br>• Initiating new linkages with and insights into the functioning of other sectors and departments for improved understanding of public services as well as of the roles and responsibilities of different stakeholders; |
|  | Information | • Community knowledge is a rich and vivid source of sophisticated information;<br>• Information that is coproduced with citizens and service providers confers legitimacy;<br>• Clear, accessible information that is useful and actionable by stakeholders is more easily integrated into routines;<br>• Connecting to the health observatories brings additional data to bear on community-nominated priorities; |
| Mechanisms | Processes, dynamics and interfaces | • Regular community engagement builds strategic, analytical and public-speaking skills and confidence;<br>• Regular engagement between communities and authorities, fosters awareness, mutual understanding and trust;<br>• 'Safe spaces' outside institutional processes to connect with and understand other agencies and communities are valuable;<br>• Spaces close to implementation contexts support inclusivity, managed expectations, reinforced principles, and a process owned and controlled by those involved;<br>• Processes framed as shared endeavours can deepen engagement, ownership and understanding;<br>• Processes need time to build and maintain constructive, cooperative relationships and trust between sectors;<br>• Monitoring is effectively done by those closest to the issue;<br>• Recognition of roles of mediators is important–facilitators between communities and authorities with two-way communication "bridging cultural and power gaps";<br>• Incentives and renumeration require careful consideration as positive reinforcement to sustain the practice; |
| Outputs/ outcomes | Legitimate process, ownership, uptake and collective action | • Identify range of intermediate and ultimate outcomes and timeframes with sensitivity to challenges in operational levels;<br>• Continually and collectively test and revise assumptions about theories of change to build relationships and trust;<br>• Document and emphasise improved engagement and mutual understanding, capacity in deliberative processes;<br>• Develop collective action towards shared priorities, joint reflection and adaption of process;<br>• Coordinate with government reforms, processes and priorities. Uptake of process into routine planning and management processes important to support state response;<br>• Sandwich strategies' building citizen voice and support state responsiveness can help build mutual respect, understanding, and ultimately mutual empowerment. Analysis should focus on different ways that 'voice' and 'teeth' combine, interact and situate in particular contexts; |

*(Continued)*

 

**Table 3.** (Continued)

| Theory of Change | Analytical domain | Transferable learning |
|---|---|---|
| Contexts | Meso/micro context (structure) | • Identify capacity built on local relationships for innovation, efficiency and responsiveness to improve the quality of service delivery;<br>• Build on existing processes to avoid imposing administrative burdens in already constrained operational environments;<br>• Sustainability key consideration: iterative, dynamic and responsive/sensitive to realities on the ground to support strategic, empowerment-focussed approaches to emerge and have legitimacy;<br>• Past experience of interaction with the state, trust, cultures of expectations from the state need to be recognised; |
| | Macro context (structure) | • Focussing only on local, front-line service providers, should be supplemented with attention to higher levels;<br>• Attention to macro-level social and political contexts can help to identify how impacts are both supported and undermined;<br>• Forward-looking, preventative approaches may be challenging to advance in contexts of limited accountability;<br>• Universal Health Coverage and Primary Health Care are important supporting reforms; however realities of implementation reflect deep tensions that require dedicated analysis;<br>• Design as longer political process of citizen engagement with the state;<br>• Consider how to improve incentives to sustainably engage higher-level leadership for policy design and implementation. |

Framed in terms of social accountability, the process thus made gains raising community *voice* and initiating multisectoral dialogue with the authorities, giving the voice *teeth*. Achieving *bite*, however, requires longer-term engagement and more formal connections to the system [74]. Ultimately, institutionalising within sectoral processes supports sustainability. Avoiding additional administrative burdens in already constrained environments is highly important. While creating spaces outside institutional environments was well-received, significant time and energy were required to maintain relationships, and multiple, competing priorities ultimately undermined DoH stakeholders' ability to progress action. In addition, financial compensation for participants may not be sustainable and consideration of incentives for long-term engagement and to engage higher-level leadership is key. While a relatively short-term research project is not likely to be commensurate with the long-term processes and partnerships required, the intention is that, through continued cycles, small-scale work can support longer-term engagement.

## Relevance and practical utility

Taking control of the agenda, community stakeholders identified an entrenched social problem conferring significant burdens on rural communities and the health system. This finding has wider relevance. While HIV/AIDS is declining in South Africa, drug and alcohol abuse is *increasing* among youth and adolescents linked to intergenerational trauma, high levels of violence and accidents, and emergency medical services of low quality [75–78]. Globally, alcohol-related diseases and injuries take approximately 3.3 million lives every year, with over 190,000 drug-related deaths recorded in 2015 [79]. Despite this, substance abuse is a low priority in low- and middle-income countries (LMICs) and evidence-based policy is lacking [80,81]. In South Africa, alcohol industry influence on regulatory reforms over sale and advertising is

 

strong [82]. Despite this, however, sales were repeatedly banned during COVID-19 lockdowns and alcohol-related harms reduced significantly: by up to 65% in trauma cases during this time [83,84] supporting renewed calls for regulatory reform [85].

In the neutral learning spaces, policy and systems responses to AOD abuse were examined candidly and in detail; with policy-implementation gaps affirmed as a critical issue. This finding also has wider relevance. While the 2013–17 National Drug Master Plan adopted a joined-up harm reduction approach [86], no reduction in AOD abuse was observed by 2016 and the absence of an implementation strategy was acknowledged in the subsequent 2019–24 plan [87,88]. There is broad recognition of the lack of attention to effective policy implementation in South Africa and across many LMICs, in public health systems characterised by underinvestment, limited links with communities, corruption, deep-rooted inequalities and political instabilities [29,89]. Regular dialogue and exchange supported attention to deeply constrained operational contexts and how these limit progress. In doing so, conducive relationships were developed between communities and the authorities, resulting in shifts towards collaborative mindsets, and recommendations to embed the process into formal PHC planning and review [67].

The process surfaced a clear operational need: to support CHWs to connect with communities and rapidly generate evidence on local needs and situations [90]. There are many obstacles to integrating CHWs into the public system, however. Implementation of WBPHCOTs has been slow and uneven and there is low coverage [43]. There is also low awareness of expanded CHW roles and functions in communities, resulting in roles that are not well-defined, valued or supported [91,92]. Nevertheless, within and beyond South Africa, CHWs have made critical contributions in local surveillance and response efforts for more informed responses, supporting calls for recognition of, and support for, this critical cadre [93–99].

In terms of practical utility, institutionalising within sectoral processes and maintaining linkages in highly fluid contexts will undoubtedly present challenges. As described above, despite broad PHC revival, operational accountability spaces are restricted in South Africa [34]. Top-down governance persists, overlooking significant ingenuity, innovation and resilience at lower levels [33,89,100]. International evidence indicates entry and leverage points, however. In the Philippines, connecting communities to PHC was supported through careful examination of roles and motivations [101]. And, in Nepal, recognition of local value and relevance of participatory action learning has supported sustained action over time [102].

Sustainability is also considered relative to challenges and opportunities related to COVID-19. South Africa had timely and decisive action in response to COVID-19 with strict lockdowns, integrated support and nationwide community-based screening and testing [103]. While a highly centralised strategy initially slowed the rise in cases, the phased lifting of lockdown has been accompanied by further waves, driven by new, more transmissible variants. There have also been severe impacts on incomes and food security, particularly in informal settlements, and there are serious concerns over diagnosis and treatment of other conditions, particularly HIV/AIDS and TB [45,104,105]. The pandemic brings new demands to already-challenged systems, nevertheless, it also underscores the necessity of real-time local data and action, community involvement, and multisectoral approaches.

Finally, exclusively 'local' accountability processes may be limited, based on assumptions that problems are only local [48,49]. Longer-term processes connecting to higher levels with power to act, and 'sandwich strategies' building citizen voice *and* supporting state responsiveness can help navigate hierarchical organisational contexts. In Thailand, for example, the National Health Assembly supports coalitions of civil society, government and academics in participatory health systems governance and is a prominent example of virtuous cycles of voice and response [106]. Social and political contexts exert considerable influence on whether

and how accountability process exist and are effective. Our analysis reflects that, a deeply hierarchical system notwithstanding, nurturing and democratising the *social processes* through which agents and structures interact and influence one another can support shifts towards cooperative mindsets, alliances, and new ways of thinking. Future analyses should focus on understanding the ways that 'voice' and 'teeth' combine, interact and situate in contexts that institutionalise participation as a rights-based approach to health [7,107,108].

## Conclusion

Despite normative support, there is limited operational understanding of *how* to progress multisectoral action on health inequalities with meaningful inclusion of underserved communities. Tensions exist between policy commitments to bring services closer to people and restricted operational spaces inclusive of community voices to understand and respond to local health priorities. Through an action-learning process to support and enable mutual accountability among communities and health systems actors, we elicited community intelligence on local health concerns, quantified associated burdens of disease, and supported stakeholders from rural communities and the authorities to build dialogue, partnerships, and to develop, implement and evaluate local action. The process supported otherwise disconnected actors to build collaborative mindsets and collectively progress and learn about action. The potential for impact was identified, with acceptability and practical utility affirmed in recommendations to embed into routine PHC planning and review. In contexts of deeply embedded social and health inequalities, nurturing meaningful citizen-state interfaces, such as those reported here via cooperative learning platforms, can support the realisation of meaningful community participation and multisectoral action.

## Supporting information

**S1 Table. Government and community stakeholder analysis of AOD abuse in rural communities.**
(DOCX)

**S2 Table. Governmental and community stakeholder analysis of existing policies, implementing partners and recommendations to address AOD abuse.**
(DOCX)

**S3 Table. Local Action Plan to address AOD abuse among youth and adolescents.**
(DOCX)

**S1 Text. Department of Health Research Brief.**
(DOCX)

**S2 Text. Local Action Plan monitoring proforma.**
(DOCX)

## Acknowledgments

The authors would like to thank community, government, and non-governmental stakeholder participants for agreeing to be part of the process, and for sharing their time, knowledge, and perspectives. Thanks also to the Verbal Autopsy with Participatory Action Research (VAPAR) team and staff of the Medical Research Council (MRC)/Wits Rural Public Health and Health Transitions Research Unit (Agincourt), especially Simon Khoza, Sizzy Ngobeni and Ella

Sihlangu. Thanks to Roosamaria Savela for help with literature searches. Finally, thanks to Rene Loewenson for comments on an earlier draft.

## Author Contributions

**Conceptualization:** Lucia D'Ambruoso, Maria van der Merwe, Sophie Witter.

**Data curation:** Lucia D'Ambruoso, Denny Mabetha, Rhian Twine, Maria van der Merwe, Jennifer Hove.

**Formal analysis:** Lucia D'Ambruoso, Denny Mabetha, Rhian Twine, Maria van der Merwe, Gerhard Goosen, Jerry Sigudla, Sophie Witter.

**Funding acquisition:** Lucia D'Ambruoso.

**Investigation:** Lucia D'Ambruoso, Denny Mabetha.

**Methodology:** Lucia D'Ambruoso, Denny Mabetha, Rhian Twine, Maria van der Merwe, Gerhard Goosen, Jerry Sigudla, Sophie Witter.

**Project administration:** Lucia D'Ambruoso, Denny Mabetha, Rhian Twine, Maria van der Merwe, Jennifer Hove, Sophie Witter.

**Resources:** Lucia D'Ambruoso.

**Supervision:** Denny Mabetha.

**Writing – original draft:** Lucia D'Ambruoso.

**Writing – review & editing:** Denny Mabetha, Rhian Twine, Maria van der Merwe, Jennifer Hove, Gerhard Goosen, Jerry Sigudla, Sophie Witter.

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
