## [Decision Letter · Decision Letter 0]

7 Feb 2022

PGPH-D-21-00979

‘Voice needs teeth to have bite’! Progressing community-led multisectoral action-learning to address alcohol and drug abuse in rural South Africa

Dear Dr. D'Ambruoso,

Thank you for submitting your manuscript to PLOS Global Public Health. After careful consideration, we feel that it has merit but does not fully meet PLOS Global Public Health’s publication criteria as it currently stands. Therefore, we invite you to submit a revised version of the manuscript that addresses the points raised during the review process.

A rebuttal letter that responds to each point raised by the reviewers. You should upload this letter as a separate file labeled 'Response to Reviewers'.A marked-up copy of your manuscript that highlights changes made to the original version. You should upload this as a separate file labeled 'Revised Manuscript with Track Changes'.An unmarked version of your revised paper without tracked changes. You should upload this as a separate file labeled 'Manuscript'.

We look forward to receiving your revised manuscript.

Kind regards,

Jose Ignacio Nazif-Munoz, Ph.D.

Academic Editor

Journal Requirements:

1. Please provide separate figure files in .tif or .eps format only.  Please ensure that all files are under our size limit of 20MB.  

For more information about how to convert your figure files please see our guidelines: Once you've converted your files to .tif or .eps, please also make sure that your figures meet our format requirements

2. Please update the completed 'Competing Interests' statement, including any COIs declared by your co-authors. If you have no competing interests to declare, please state "The authors have declared that no competing interests exist".

3. We have noticed that you have uploaded supporting information but you have not included a list of legends.  Please add a full list of legends for all supporting information files (including figures, table and data files) after the references list. 

4. In the online submission form, you indicated that "The data are available from the authors on reasonable request.". All PLOS journals now require all data underlying the findings described in their manuscript to be freely available to other researchers, either 1. In a public repository, 2. Within the manuscript itself, or 3. Uploaded as supplementary information.

5. Please amend your detailed Financial Disclosure statement. This is published with the article, therefore should be completed in full sentences and contain the exact wording you wish to be published.

i) Please include all sources of funding (financial or material support) for your study. List the grants (with grant number) or organizations (with url) that supported your study, including funding received from your institution. 

ii). State the initials, alongside each funding source, of each author to receive each grant.

iii). State what role the funders took in the study. If the funders had no role in your study, please state: “The funders had no role in study design, data collection and analysis, decision to publish, or preparation of the manuscript.”

Reviewers' comments:

Reviewer's Responses to Questions

**Comments to the Author**

1. Does this manuscript meet PLOS Global Public Health’s publication criteria? Is the manuscript technically sound, and do the data support the conclusions? The manuscript must describe methodologically and ethically rigorous research with conclusions that are appropriately drawn based on the data presented.

Reviewer #1: Yes

Reviewer #2: Yes

2. Has the statistical analysis been performed appropriately and rigorously?

Reviewer #1: N/A

Reviewer #2: N/A

3. Have the authors made all data underlying the findings in their manuscript fully available (please refer to the Data Availability Statement at the start of the manuscript PDF file)?

Reviewer #1: No

Reviewer #2: No

4. Is the manuscript presented in an intelligible fashion and written in standard English?

Reviewer #1: Yes

Reviewer #2: Yes

5. Review Comments to the Author

Reviewer #1: The topic and research process are very interesting, being a good example of participative process to solve health problems highlighting the normal limitations of this type of initiatives.

I provide some suggestions mostly focus to improve the presentation of the manuscript

Results

While reading the results are difficult to follow. The researchers mention as themes:

a) dynamics of the process.

b) actor interactions and interfaces.

c) impacts and degrees of impacts.

d) enabling and constraining contextual factors.

Nonetheless, the results do not highlight the above, presenting other subtitles that promote confusion. Also, the results do not include quotes or evidence of the description presented, the researchers present observation of actor’s dynamics, combined with impressions and actions of the research team and theoretical arguments of the methodology. I suggest revise and restructure this section.

I suggest including:

- A summary of the findings from the VA to understand the magnitude of the problem in the manuscript going beyond the 30% of prevalence of the problem. This allows a better understanding of the resources considered by the different actors.

- A table with the principal findings related to PAR in every phase.

- A Figure (or more if is needed) that represent the real action-learning process that resulted from the implementation of PAR. This allows to respond in a clear way to the main objective of the research.

Discussion

“Derive transferrable learning” is the second objective, I suggest moving it from the discussion to the results section.

If the quote placed in the discussion is not really needed, I suggest to eliminate.

I suggest adding a paragraph that present the discussion points that will be develop in this section, I encourage authors to present it in a structured way for a more friendly reading.

References

42% of the reference have more than 5 years. I suggest updating some of them.

Reviewer #2: Article: “Voice needs teeth to have bite! Progressing community-led multisectoral action-learning to address alcohol and drug abuse in rural South Africa”

This paper opens a window into the experience of drawing together rural community stakeholders to examine and address the local public health crisis of AOD deaths in South Africa via action-learning processes with diligence and honesty. Processes are recounted systematically so that the reader feels immersed in the many, well-planned research stages and can understand the strategies and outcome goals embedded at every stage of presentation.

Clearly the research team’s long experience in participatory methods and study design come through with ease so that the reader can derive much from this well-written formal accounting of the intervention.

My comments are therefore offered to expand the context of study particulars study so as to distinguish it from any comparable action-learning experience. Because the researchers are so immersed, and so knowledgeable of the rural community public health field--its structures (internal and vis-à-vis local and national governing authorities), culture, deficiencies, and on-going resource challenges—the paper overall assumes conveys a very process-oriented tone, though at the risk of losing a sense of societal particulars which make the AOD crisis both personal and poignant within South Africa’s countryside.

This problem of context can be easily resolved through the inclusion of more anonymized participant quotes. As it stands, only three quotes are presented, and one at the very end rather than in the results section where one expects to find them. Participant quotes are the heart of qualitative thematic data and convey what figures and prose cannot.

A second related issue is that the dimensions of the AOD crisis are described late in the paper as part of the discussion. In fact, their appropriateness is wanting in the paper’s Introduction to set the tone of urgency currently missing as one reads sections of recruitment, report drafting, and analysis so meticulously described. As is, it feels like there is an excessive detachment from the underlying health problem whose recognition is being process-driven.

Nonetheless, I recommend this paper for publication with minor revisions described in detail below.

Thank you for this opportunity to review this paper, which I greatly appreciated.

Abstract: Really good abstract, with clear goals, methods, findings & nice recap of conclusions as separate to discussion & injecting timely reflections from vantage point of participatory action-learning facilitators.

Intro: Please provide some background to drug and alcohol abuse generally in South Africa and why, other than the misclassification of deaths due to drug/alcohol abuse. Statistics are provided for HIV, but then you apply the methods to a different health scourge altogether, and this seems like an oversight as it stands. LNS 440-445 provide this information, and while not inappropriate for discussion, feel like it would set the scene better in intro for researchers’ choice to focus on AOD for PAR & not other issue.

VAPAR, even though VAPAR)/Wits/Mpumalanga/Department of Health Learning Platform as an organizational collaborator on the research (as per title page), and approach explained on LN 136, there is still a disconnect between the reported causes of death and its application to alcohol & drug-related behaviors. This term is unfamiliar, even though it is a key word. Please make connect of VAPAR to PAR, as it remains unclear.

LN 69 use of in-action: a play on words (i.e., inaction), or a subset like action-learning? Not being familiar with sector terminology, this term created confusion.

LN 82: HIV prevalence…missing the word IS in 1st part of sentence

LN 86: human resources crises related to pandemic, exacerbated by it, or previous to pandemic as well? Maybe return to this point & connect it to paragraph beginning LN 105.

LN 153: policy implementation gaps—situate this statement in the context of official post-apartheid South Africa. Is health sector any better or worse than other sectors? And give some framing of why local community health delivery is more promising to close those policy gaps than other sectors.

Recruitment and snowballing of interviewees as study population grew is very well described. One place to expand is LN 172: what is population of village & size of regional area of local community-based health care ward? Having context to better understand the overall challenge you are stating would be well served by problem context. Literacy rates of community? Unemployment rates among young people affected by alcohol & drug abuse. Reader is left to conjecture causes of problem without facts.

Another is at LN 174: How was recruitment of initial participants undertaken, by reputation, by nomination of another health organization? Please provide brief clarification as to how they were selected. LNs 183-184: same issue for next stage of nominations.

Nice sequence of meeting steps under Step 2: Analyse/Plan.

LN 206: need “on” at end of line

LN 234: copy of “structured tool” in supplementary materials would be helpful to see.

LN 510: Local Action Plan: which stakeholders involved, at what levels of authority. This is necessary before

dissection of its drawbacks & potential challenges to implementation.

LN 239: 16 follow-up visits to whom? At this point, it needs to be reestablished who your active participants are. Would be helpful to summarize prior to analysis how many total individuals overall participated in the process and contributed to your collected data?

LNs 260 & 261: Structuration theory. Both lines support your public health research use of the theory for institutional-actor interrelations. But for those unfamiliar with this technical term, would be useful to define it here independently of the many citations.

Methods: did you use any qualitative data software or organizing system for thematic coding & analysis? Please specify software name if you did.

Some more knowledge of participant base: different ethnicities/groups represented, any division along lines of authority based on tribe-based financial status or privilege? In other words, was the arena for discussion “flat” in terms of power-bases?

Similarly, at LN 523: South Africa’s COVID response was noteworthy, but in which direction? Did community-level implementation trump the competing challenges you mention, or did this impede success? Before moving on to future urgency, need to inform how successful (and therefore imitable) COVID community-level response was.

Need more quotes such as LN 565 by Joshi. Very informative to hear in first-person voice participant reactions & recognitions of reality behind social accountability and perceived effectiveness of citizen engagement.

Methodological reflections are very strong, thoughtfully presented and grounded in research team’s long-term engagement and expertise in rural community health. While transferal to subnational and national contexts is raised, one would be curious as to applicability to extra-national contexts? Is success of the PAR approach contingent on civil society functionality? Thailand is mentioned, where trust is/was high; South Africa, one would suspect, less so, given histories of two contrasted countries. A note on generalizability in light of political cultures where PAR is being pursued is therefore merited.

Do not feel that figures 1 or 2 add to reader’s understanding of the process, perhaps due to the quality of the reproduction or the density of the text in these figures and suggest dropping them. The three tables are excellent in their organization and content, and the photos capturing participant interaction very descriptive without need of these additional visuals. I feel in the end, they draw away from the effectiveness of your other visuals the research team furnishes.

6. PLOS authors have the option to publish the peer review history of their article (what does this mean?). If published, this will include your full peer review and any attached files.

**Do you want your identity to be public for this peer review?** For information about this choice, including consent withdrawal, please see our Privacy Policy.

Reviewer #1: **Yes: **

Reviewer #2: No

---

## [Decision Letter · Decision Letter 1]

21 Jun 2022

PGPH-D-21-00979R1

‘Voice needs teeth to have bite’! Expanding community-led multisectoral action-learning to address alcohol and drug abuse in rural South Africa

Dear Dr. D'Ambruoso ,

Thank you for submitting your manuscript to PLOS Global Public Health. We invite you to submit a revised version of the manuscript that addresses the points raised regarding your discussion according to reviewer 1, and minor edits from reviewer 2.

We look forward to receiving your revised manuscript.

Kind regards,

Jose Ignacio Nazif-Munoz, Ph.D.

Academic Editor

Journal Requirements:

Reviewers' comments:

Reviewer's Responses to Questions

**Comments to the Author**

1. If the authors have adequately addressed your comments raised in a previous round of review and you feel that this manuscript is now acceptable for publication, you may indicate that here to bypass the “Comments to the Author” section, enter your conflict of interest statement in the “Confidential to Editor” section, and submit your "Accept" recommendation.

Reviewer #1: All comments have been addressed

Reviewer #2: All comments have been addressed

2. Does this manuscript meet PLOS Global Public Health’s publication criteria? Is the manuscript technically sound, and do the data support the conclusions? The manuscript must describe methodologically and ethically rigorous research with conclusions that are appropriately drawn based on the data presented.

Reviewer #1: Yes

Reviewer #2: Yes

3. Has the statistical analysis been performed appropriately and rigorously?

Reviewer #1: N/A

Reviewer #2: N/A

4. Have the authors made all data underlying the findings in their manuscript fully available (please refer to the Data Availability Statement at the start of the manuscript PDF file)?

Reviewer #1: Yes

Reviewer #2: Yes

5. Is the manuscript presented in an intelligible fashion and written in standard English?

Reviewer #1: Yes

Reviewer #2: Yes

6. Review Comments to the Author

Reviewer #1: I appreciate the opportunity to review this manuscript.

I only have comments regarding the discussion.

I suggest restructuring this by pointing to the results obtained that are related to the participatory process (process forms and dynamics; actor interactions and interfaces; impacts and degrees of impacts; and enabling and constraining contextual factors) rather than the issue of AOD.

The way in which the discussion is currently structured seems confusing to me, opening spaces that until then had not been presented (such as COVID) and that may be part of the limitations of the study. In particular, I suggest generating one full reflective argumentative paragraph about the level of achievement of the original plan and another that addresses this point "...giving the voice teeth. Achieving bite, however, requires longer-term engagement and more formal connections to the system." mainly because it happens to be the title of the manuscript.

Reviewer #2: Dear authors, Congratulations on returning such a strong revision of your important initial submission. You thoughtfully incorporated both reviewers’ comments and suggestions, making for a strongly grounded and fluently argued case study of VAPAR. The added introductory Detail of South Africa’s AOD’s landscape provides the necessary and compelling urgency of the problem you later to explore. Technical terms like VAPAR and structuralization are satisfactorily explained. Participant quotes are vivid and well-placed lending in an immediacy missing prior.

Ver minor details to attend to which do not hold up my recommendation for publication, but please address:

1.line 377-78: missing the word are after “something you”

2. Line 405: “otherwise… (Supplementary information)”. Something dropped

3. Line 427: [The workshops] . Drop the it at beginning of sentence

4. Line 486: some (xx). What you have in parentheses currently modifies some

5. Line 510: eliminate colon after to

6. Line 515: The section at end of results titled transferable learning seems more appropriate for the discussion section

7. PLOS authors have the option to publish the peer review history of their article (what does this mean?). If published, this will include your full peer review and any attached files.

**Do you want your identity to be public for this peer review?** For information about this choice, including consent withdrawal, please see our Privacy Policy.

Reviewer #1: **Yes: **Karen A. Dominguez-Cancino

Reviewer #2: No

---

## [Editor Report · Decision Letter 2]

5 Jul 2022

‘Voice needs teeth to have bite’! Expanding community-led multisectoral action-learning to address alcohol and drug abuse in rural South Africa

PGPH-D-21-00979R2

Dear Dr Ambrosio,

We are pleased to inform you that your manuscript '‘Voice needs teeth to have bite’! Expanding community-led multisectoral action-learning to address alcohol and drug abuse in rural South Africa' has been provisionally accepted for publication in PLOS Global Public Health.

Best regards,

Jose Ignacio Nazif-Munoz, Ph.D.

Academic Editor